

# Influence of natural and anthropogenic aerosol on cloud base droplet size distributions in clouds over the South China Sea and Western Pacific

Rose M. Miller[1], Robert M. Rauber[1], Larry Di Girolamo[1], Matthew Rilloraza[1], Dongwei Fu[1,2], Greg M. McFarquhar[3,4], Stephen W. Nesbitt[1], Luke D. Ziemba[5], Sarah Woods[6], K. Lee Thornhill[5,7]

[1] Department of Atmospheric Science, University of Illinois Urbana-Champaign, Urbana, IL, USA

[2] Space Science and Engineering Center, University of Wisconsin-Madison, Madison, WI, USA

[3] Cooperative Institute of Severe and High Impact Weather Research and Operations, University of Oklahoma, Norman, OK, USA

[4] School of Meteorology, University of Oklahoma, OK, USA

[5] NASA Langley Research Center, Hampton, VA, USA

[6] National Center for Atmospheric Research, Boulder, CO, USA

[7] Science Systems and Applications, Inc., Hampton, VA, USA

*Correspondence to:* Rose M. Miller (rosemm2@illinois.edu)



**Abstract.** Cumulus clouds are common over maritime regions. They are important regulators of the global radiative energy budget and global hydrologic cycle, and a key contributor to the uncertainty in anthropogenic climate change projections due to uncertainty in aerosol-cloud interactions. These interactions are regionally specific owing to their strong influences on aerosol sources and meteorology. Here, our analysis focuses on the statistical properties of marine boundary layer (MBL) aerosol chemistry and the relationships of MBL aerosol to cumulus cloud properties just above cloud base as sampled in 2019 during the NASA Cloud, Aerosol and Monsoon Processes Philippines Experiment (CAMP$^2$Ex). The aerosol and clouds were sampled by instruments on the NASA P-3 aircraft over three distinct maritime regions around the Philippines: the West Pacific, the South China Sea, and the Sulu Sea.

Our analysis show three primary sources influenced the aerosol chemical composition: marine (ocean source), industrial (Southeast Asia, Manila, and cargo and tanker ship emissions), and biomass burning (Borneo and Indonesia). The marine aerosol chemical composition had low values of all sampled chemical signatures, specifically median values of 2.3 µg/m$^3$ of organics (ORG), 6.1 µg/m$^3$ of SO$_4$, 0.1 µg/m$^3$ of NO$_3$, 1.4 µg/m$^3$ of NH$_4$, 0.04 µg/m$^3$ of Cl, and 0.0074 µg/m$^3$ of refractory black carbon (BC). Chemical signatures of the other two aerosol source regions were: industrial, with elevated SO$_4$ having a median value of 6.1 µg/m$^3$ and biomass burning, with elevated median concentrations of ORG 21.2 µg/m$^3$ and BC 0.1351 µg/m$^3$. The industrial component was primarily from ship emissions based on chemical signatures. The ship emissions were sampled within 60 km of ships and within projected ship plumes. Normalized cloud-droplet size distributions in clouds sampled near the MBL passes of the P-3 showed that clouds impacted by industrial and biomass burning contained higher concentrations of cloud droplets, by as much as 1.5 orders of magnitude for sizes with diameters < 13 µm compared to marine clouds, while at size ranges between 13.0 - 34.5 µm the median concentrations of cloud droplets in all aerosol categories were nearly an order of magnitude less than the marine category. In the droplet size bins centered at diameters > 34.5 µm concentrations were equal to, or slightly exceeded, the concentrations of the marine clouds. These analyses show that anthropogenic aerosol generated from industrial and biomass burning sources significantly influence cloud base microphysical



structure in the Philippine region enhancing the small droplet concentration and reducing the concentration of mid-sized droplets.



## 1 Introduction


Aerosol and cloud interactions have long been one of the largest uncertainties in anthropogenic climate change predictions (IPCC, 2021). Efforts to intensify aerosol-cloud interaction research aimed at specific regions has been called for (e.g., Stevens and Feingold, 2009) to understand their responses to different aerosol sources and environmental conditions. Southeast

Asia is a quintessential research location to investigate a variety of aerosol emissions and their subsequent impact on tropical clouds (Reid et al., 2013, 2015). Biomass burning (BB) aerosol in the Southeast Asia region, which result from fires that are both natural and anthropogenic, have both a direct and semi-direct radiative effect (e.g. Lin et al, 2014; Ding et al., 2021; Mallet et al., 2021). BB aerosol absorb and scatter solar radiation that affects the lifetime and properties of

clouds (e.g. Andreae, 1991; Penner et al., 1992; Ackerman et al., 2000a; Bond et al., 2013) and influence regional and global climate (Crutzen and Andreae, 1990). BB aerosol also impact cloud condensation nuclei (CCN) concentrations, their activation, and droplet formation (Kacarab et al., 2020; Zheng et al., 2020). In the Southeast Asia region, the semi-direct effect of BB aerosol in the vertical direction intensifies low cloud cover over ocean and land (Ding et al., 2021).


Other aerosol produced in this region result from both anthropogenic and natural sources. Natural aerosol include sea salt, and mineral dust, amongst others, while anthropogenic aerosol are dominated by organics, sulfates, black carbon (BC), and nitrates. BC aerosol are formed from the incomplete combustion of hydrocarbons, e.g., coal power plants, agricultural BB, and combustion engines (Zhang et al., 2012; Li et al., 2016), with primary sources in the large urban

areas of Southeast Asia. Long-range southeastward transport of anthropogenic aerosol from East Asia have been measured in the South China Sea (Wang et al., 2013; Lin et al., 2014). Additionally, Manila urban pollution has exceedingly high BC concentrations from diesel exhaust (Bond and Bergstrom, 2006). Nitrate aerosol scatter radiation more effectively and their concentrations in the atmosphere may surpass sulfate levels in the near future (An et al., 2019; Zhang et al., 2012). The

impact of anthropogenic aerosol such as sulfate, nitrate, and BC has been a main topic of interest for many years as they lead to an increase of CCN that increases the cloud droplet number concentration ($N_d$) and decreases the effective radius ($r_e$) of the droplets, producing more reflective clouds for the same liquid water path (e.g. Twomey, 1974, 1977; Ackerman et al., 2000a). During a field campaign over the Indian Ocean in 1999, clouds impacted by anthropogenic aerosol had



cloud droplet concentrations up to three times greater than in clean marine clouds, along with an increase in cloud optical depth (Heymsfield and McFarquhar, 2001; McFarqhuar et al., 2004).

Shipping and marine traffic also introduces aerosol over marine areas, particularly near shipping lanes (Marmer and Langmann, 2005). In terms of anthropogenic aerosol, shipping pollution is the largest and least regulated source of anthropogenic pollutants over oceans (Marmer

and Langmann, 2005), emitting carbon monoxide (CO), sulfur oxides (SOx), nitrogen oxides (NOx), particulate matter (PM), volatile organic compounds (VOCs) and greenhouse gases into the atmosphere on a constant basis (Corbett and Fischbeck, 1997). Ship tracks, and shipping emissions from individual ships, have been studied since the 1960s (Eyring et al., 2005). Shipping is expected to contribute to 17% of global $CO_2$ emissions in 2050 (Cames et al., 2015). The impact

of shipping pollution on marine clouds and precipitation has been explored in recent decades (Petzold et al., 2008, Rosenfeld et al., 2008; Stevens and Feingold, 2009; Coggon et al., 2012; Juwono et al., 2013; Russell et al., 2013; Gryspeerdt et al., 2019; Toll et al., 2019; Manshausen et al., 2022). For example, Radke et al., (1989) observed an increase in total cloud droplet concentrations, but a decrease in cloud droplet sizes in clouds over shipping lanes. Cloud droplet

number has also been reported to increase with aerosol loading over the East China Sea (Bennartz et al., 2011). Ships emit carbonaceous particles from burning fuel. They also produce sulfur dioxide and sulfates that lead to increased cloud condensation nuclei CCN (Capaldo et al., 1999; Hobbs et al., 2000).

Previous field campaigns in the Southeast Asia region, such as the Seven South East Asian

Studies (7SEAS) (Reid et al., 2013), were aimed at understanding aerosol radiative effects and aerosol particle characteristics using ground and ship-based measurements (Reid et al., 2015, 2016; Hilario et al., 2020b). The impact of aerosols on low clouds in this region has been difficult to observe from satellite due to heavy cirrus cloud cover (Reid et al., 2013; Hong and Di Girolamo, 2020) and to model because of our current poor understanding of cloud properties in lower-level

clouds beneath the cirrus (Wang et al., 2013; Xian et al., 2013). Past studies of aerosol and cloud properties on aircraft-based platforms in the Southeast Asia region include the Indian Ocean Experiment (INDOEX) (Ramanathan et al., 2002), the Atmospheric Chemistry Experiment in Asia (ACE-Asia) (Huebert et al., 2003), the Atmospheric Brown Clouds project (Ramanathan et al.,



2005; Nakajima et al., 2007), and the East Asian Study of Tropospheric Aerosols: an International
Regional Experiment (EAST-AIRE) (Z. Li et al., 2011).

        During late August through October 2019, the Cloud, Aerosol and Monsoon Processes
Philippines Experiment (CAMP²Ex), which operated out of Clark International Airport on Luzon
Island in the Philippines, offered an opportunity to conduct airborne sampling in tropical maritime
convective environments that is closer to the Philippines, employing an extensive suite of aerosol,
cloud and radiation measurements. CAMP²Ex used two research aircrafts, namely the National
Aeronautics and Space Administration (NASA) P-3 and the Stratton Park Engineering Company
(SPEC), Inc. LearJet 35 to sample aerosol from three different sources, marine, BB, and industrial,
to sample the clouds influenced by these aerosol.  The Learjet 35 was also heavily instrumented
with cloud particle probes. It primarily sampled higher regions of the clouds, and thus the
observations in this paper will focus only on that collected by the P-3.

        The Southeast Asia regional meteorological and climate features, described in Reid et al.,
(2013), are key factors for aerosol transport and cloud formation and propagation throughout the
region. Large scale features include circulations such as those associated with the Southern
Oscillation (Rasmusson and Wallace, 1983; Mcbride et al., 2003) and monsoonal flows tied to
seasonal shifts in the Intertropical Convergence Zone (ITCZ) (Chang et al., 2005a; Wang et al.,
2009). Smaller scale meteorological features affecting aerosol transport and clouds include tropical
cyclones (Yasunaga et al., 2003,  Zhang et al., 2003), land/sea breezes, shallow to moderate
convection typical of fair weather in trade wind regions (e.g. Schafer et al., 2001, Zuidema et al.,
2012).

Three aerosol source regions influence the boundary layer air over the Philippine region
(Fig. 1). When the southwest monsoon flow is present, BB aerosol are advected northward over
the Sulu Sea south of Luzon from regions in Malaysia and Indonesia (Xian et al., 2013). These
regions are prone to peatland fires and human-caused agricultural fires which are enhanced during
periods of drought and El Niño conditions (Reid et al., 2012; Yin et al., 2020). Long range
southeastward transport of anthropogenic aerosol from large cities of East Asia into the South
China Sea can be present year around (Wang et al., 2013, Lin et al., 2014). Also several
international shipping lanes transect the South China Sea and Sulu Sea. All of these aerosol



combine with natural marine aerosol to produce the characteristic aerosol populations found in the oceanic boundary layer regions surrounding the Philippines.

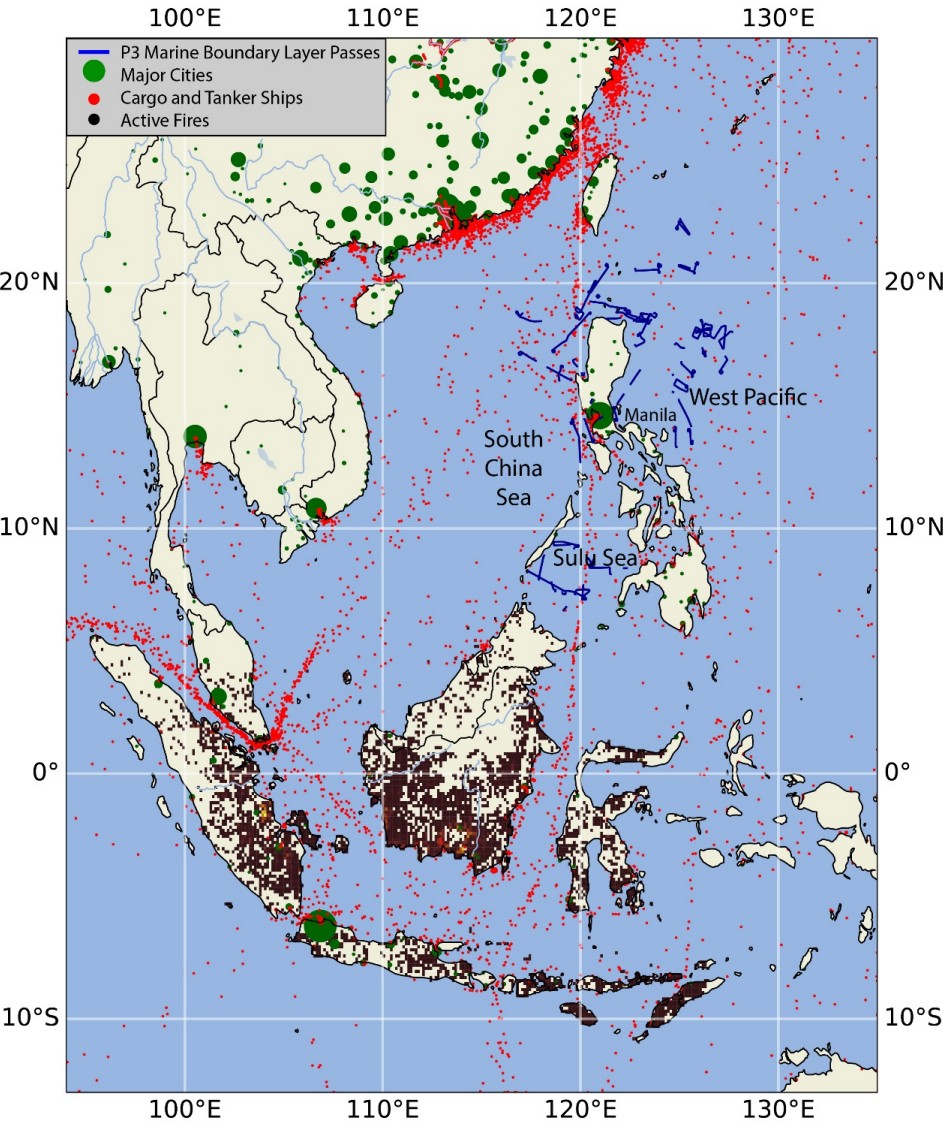


**Figure 1:** Overview map and location of the Cloud, Aerosol and Monsoon Processes Philippines Experiment campaign based out of Clark International Airport in central Luzon, Philippines. Blue lines indicate all marine boundary layer flight legs from 19 research flights from the P-3. Active fires from Fire Information for Resource Management System MODIS 6 (black dots) and cargo



and tank ship locations (red dots) are from 19 September 2019. Location of major cities with populations over one million near the sampling area (green dots) with larger dots indicating larger populations.

Herein, data from the CAMP²Ex campaign are used to determine the chemical composition of aerosol over the Philippine region from each of these three sources. Observational data of

aerosol sampled over the ocean in the MBL are categorized into clean marine aerosol, ship emissions and aged and fresh industrial pollution from mainland East Asia and Manila, and BB aerosol. The impacts of aerosol and chemical compositional differences on warm tropical cumulus clouds are then analyzed just above cloud base over the sampling region. This paper then examines how each of these aerosol types influence droplet size distributions in tropical maritime cumulus

clouds just above cloud base.

## 2 Methodology

### 2.1 CAMP²Ex

The CAMP²Ex campaign, based out of Clark International Airport, Philippines from 24 August – 5 October 2019 with a sampling region around the Philippine Islands, was designed to

characterize aerosol composition, optical and radiative properties, and their role in modulating precipitation during the Southwest Monsoon and fall transition period.  The NASA P-3 aircraft conducted 19 research flights with a payload of in-situ and remote sensing instrumentation. Here we focus on MBL passes of the P-3 aircraft used for characterization of aerosol chemical composition in the MBL, and aircraft passes sampling maritime convective clouds just above cloud

base (Fig. 2).

### 2.2 Aerosol and Cloud Physics Instrumentation

*Aerosol Clarke  inlet:* The P-3 aerosol Clarke inlet is a forward-facing shrouded solid-diffuser that is operated isokinetically that limits in-situ sampling to particles with aerodynamic diameters less than 5.0 µm (McNaughton et al., 2007). This inlet supplied sample flow to the

Aerosol Mass Spectrometer and Single-Particle Soot Photometer. All aerosol concentrations are reported at standard temperature and pressure and have been screened to remove cloud artifacts.



*Aerosol Mass Spectrometer:* The Aerodyne Time-of-Flight Aerosol Mass Spectrometer (AMS, Aerodyne Research Inc.), operated by the Langley Aerosol Research Group, was used to determine non-refractory submicron aerosol composition within aerosol plumes (Jayne et al., 2000; DeCarlo et al., 2006; Shank et al., 2012; Howell et al., 2014, Hilario et al., 2021). AMS data were used to quantitatively determine aerosol mass composition within the MBL and to classify aerosol regimes at 30-second resolution for sizes < 1 µm.

*Single Particle Soot Spectrometer:* A single particle soot spectrometer (SP2; Droplet Measurement Technologies) was used to detect refractory black carbon (rBC). The SP2 detects individual rBC particles through laser-induced incandescence (Schwarz et al., 2006; Moteki and Kondo, 2007). Black carbon is emitted through incomplete combustion processes and is used as a conserved tracer for anthropogenic aerosol sources and biomass burning emissions (Bond et al., 2013).



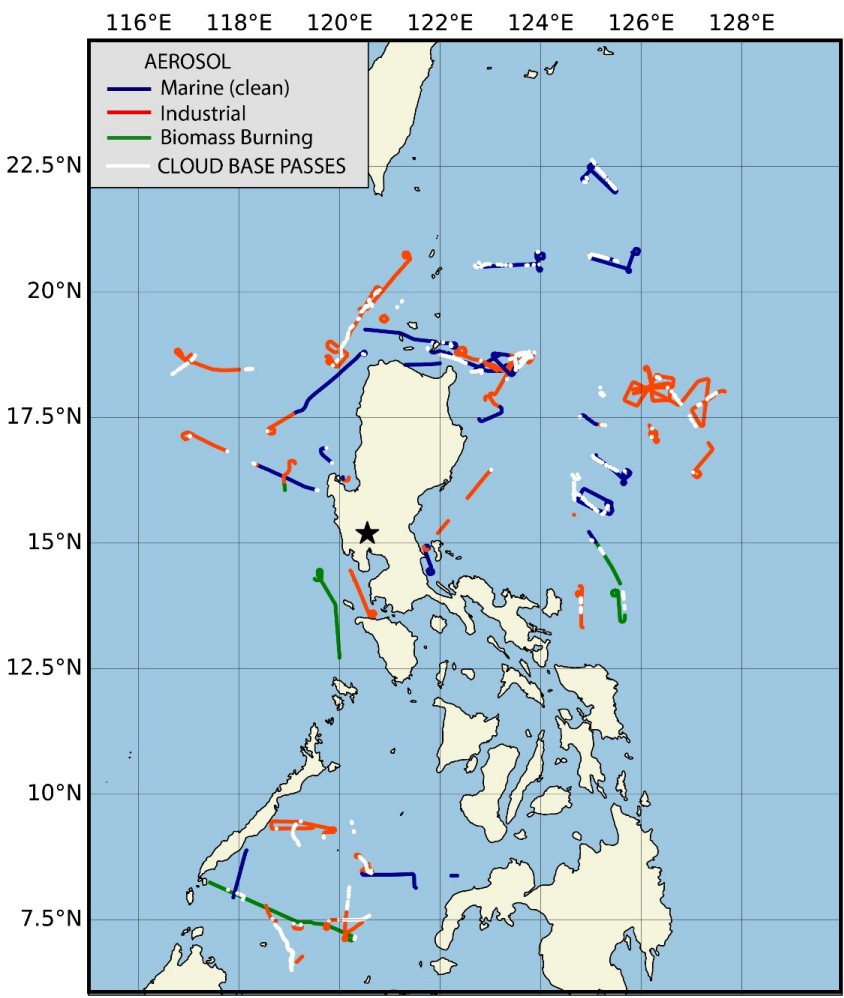

**Figure 2:** Location of all marine boundary passes (colors) from 24 August 2019 - 5 October 2019 in accordance with their assigned aerosol source region. Cloud base passes (white) are shown for all P-3 research flights where cloud base sampling occurred.

*Turbulent Air Motion Measurement System:* 3-D winds were measured from the aircrafts' attitude, position, velocity, pressure, and acceleration using the Turbulent Air Motion Measurement System (TAMMS) instrumentation from flow-angle and temperature sensors, inertial navigation and flight management systems. These measurements were made using a Rosemount Model 102 (Lenschow, 1986; Barrick et al., 1996; Thornhill et al., 2003) and derived measurements of wind components $u$, $v$, and $w$ were calculated at 20 Hz resolution.



*Fast Cloud Droplet Probe:* The Fast Cloud Droplet Probe (FCDP) SPEC Model FCDP-
100 is a forward scattering probe that measures cloud hydrometeor concentration size distributions
from 2 to 50 µm in particle diameter at 1-3 µm resolution at 1 Hz frequency (O'Connor et al.,
2008). This instrument was used to collect cloud droplet number concentrations and size
distributions above cloud base.

### 2.3 Dropsondes

During CAMP$^2$Ex 197 Vaisala RD41 dropsondes were successfully launched from the P-
3 using the Airborne Vertical Atmospheric Profiling System operated by Colorado State
University (Vömel H. et al., 2020). The data from the dropsondes were used to determine the lifting
condensation level (LCL). To determine the LCL and the vertical extent of the MBL, the nearest
dropsonde in time and space to each MBL flight leg was used, provided that the dropsonde sampled
an undisturbed environment, void of known cold pools or rain shafts. After eliminating dropsondes
that sampled disturbed environments, a Rosner's Outlier Test confirmed two outliers in the
remaining dropsonde dataset. These dropsondes were also removed. The height of the LCL was
calculated for each of the remaining 181 dropsondes. The height of the LCL for all dropsondes
was 466 ± 89 m. A distribution of the calculated heights of the LCL for all remaining dropsondes
is shown in figure 3A.



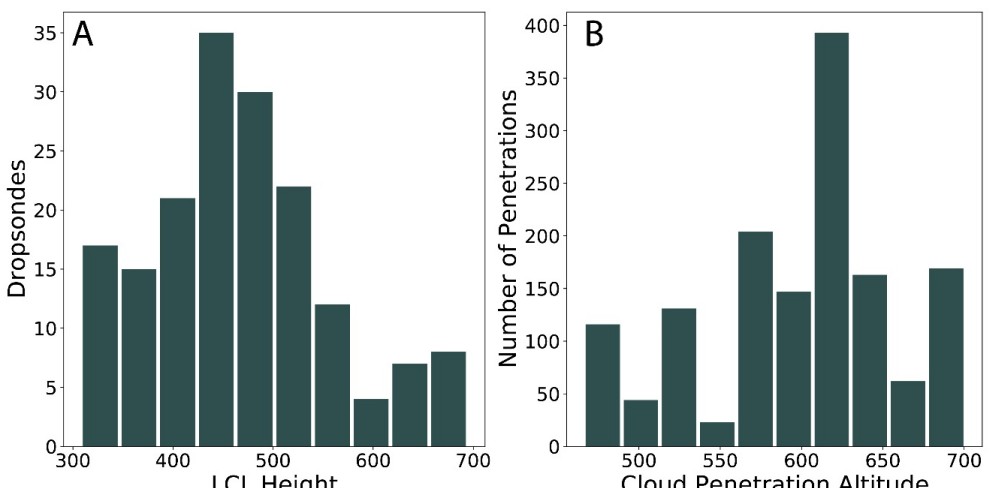

**Figure 3:** (A) Distribution of lifting condensation level heights determined from CAMP$^2$Ex

dropsondes; (B) cloud penetration altitudes of the P-3 near cloud base.

**2.4 Ship locations and ship plume trajectories**

A dataset was constructed to predict the locations of cargo and tanker ship emissions in the

vicinity of the P-3 during CAMP$^2$Ex (for methods see appendix). The dataset provided information

on the P-3 MBL status, the distance from Manila, the number of ships within a 60 km and 100 km

radius of the P-3, the number of discrete ship plumes within 60 km and 100 km of the P-3, the time

of a plume-aircraft intersection (if such an intersection occurred), the age of the intersected plume,

and the maritime mobile service identity (MMSI) location of the ship that produced that plume. A

video of ship plume and P-3 locations through each of the flight periods in included as a

supplement to this article. Lv et al. (2018) indicate that shipping emissions measured within 22 km

of a ship were normally the dominant contributor to PM2.5 aerosol. They found that shipping

emissions could be detected within 370 km of ships and shipping lanes along the China coastline.

The MMSI ship data purchased from Astra Paging Ltd provided ship information covering the

region of flights around the Philippines at 3 hr frequency between the hours of 22:00:00 UTC and

9:00:00 UTC the next day. Wind data from the ERA5 reanalysis at 1000 hPa was used to calculate

aerosol plumes produced by each ship every 600 seconds (Fig. 4, see also appendix).



## 2.5 Air parcel trajectories

NOAA's HYSPLIT model, January 2017 revision (854) version 4 (Draxler and Hess 1998, Stein et al., 2015) was used to calculate air parcel backward trajectories to determine air mass source regions during CAMP$^2$Ex. The HYSPLIT model was initialized with the Global Data Assimilation System (GDAS) at a 0.25° grid spacing. For every 10 min of each MBL sampling leg, HYSPLIT backward trajectories were calculated to estimate the origin of the air parcels. Backward trajectories from all MBL locations were initialized at or below 466 m and ran 100 hr. The trajectories were used to determine the possible location of air parcels and establish source relationships between the different aerosol source regions and the cloud base passes.




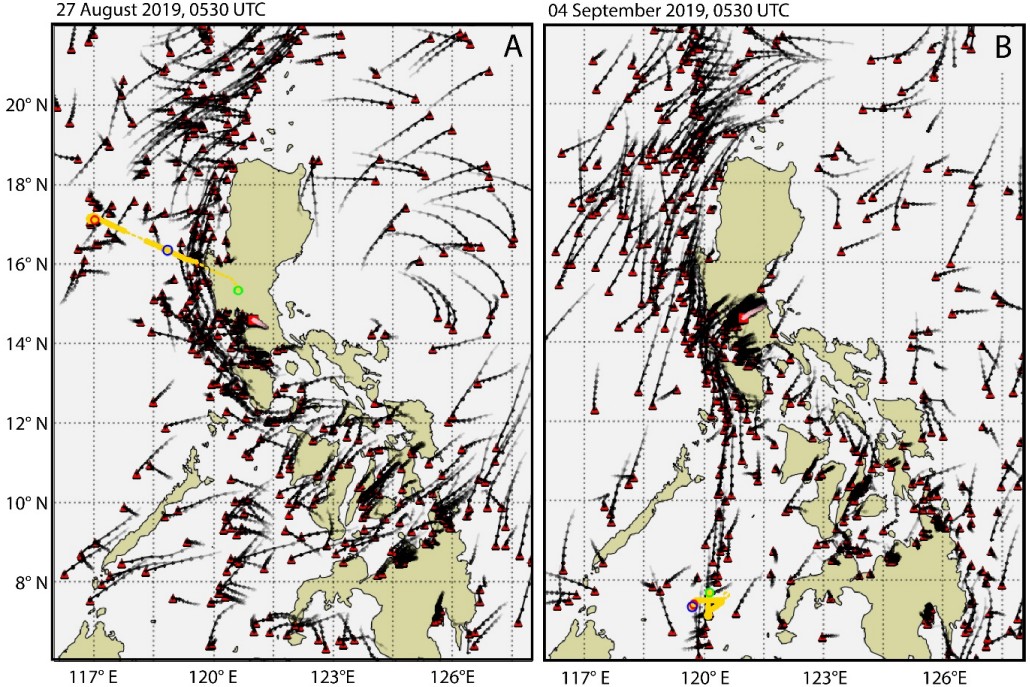

**Figure 4:** Example of ship emission projections based on European Centre for Medium-Range Weather Reanalysis Forecasts ERA5 1000 hPa winds on (A) 27 August 2019 and (B) 4 September 2019 at 05:30:00 UTC, along with flight segments of the P-3 research aircraft (yellow). Individual

cargo and tanker ships are denoted as red triangles and projected ship plumes at 30-min intervals are shown as black lines and dots over the sampling region. On both days the red ring is the P-3 research aircraft location at 05:00:00 UTC, the blue ring at 05:30:00 UTC, and the green ring at 06:00:00 UTC. The red square denotes the city of Manila and its corresponding projected pollution plume (red with white interior) is for the same period as the ship emission plumes. The thickness

of the yellow line denoted when the P-3 research aircraft is at an altitude of < 466 m (thick) and > 466 m (dashed).

## 2.6 Flight strategy

All CAMP2Ex flights were conducted during daytime between 0000 UTC – 0900 UTC. Sampling of the MBL occurred during segments of CAMP$^2$Ex flights below cloud base. All MBL



measurements reported in this paper occurred below 466 m above mean sea level (MSL), the
       median height of the LCL. The passes were divided into 10 min intervals to ensure sufficient
       sampling of the chemical species. Passes or segments of passes shorter than 10 min were not
       included in this study. Cloud sampling reported in this paper was conducted above cloud base in
       cumulus clouds below an altitude of 700 m at 1 Hz resolution (Fig. 3B).  A cloud base pass was

recorded if the FCDP reported cloud number droplet concentrations ($N_D$) $> 10 \, \mathrm{cm}^{-3}$ and liquid
       water content (LWC) $> 0.05 \, \mathrm{g \, m}^{-3}$. A total of 112 MBL passes and 1416 cloud base passes were
       recorded.

## 3 Composite Aerosol Chemical Signatures

       To identify the chemical signatures of the three distinct aerosol source regions discussed

in Sec. 1, MBL passes of the P-3 in regions with a high likelihood of having those chemical
       signatures were identified. The chemical signatures used were collected from the AMS and SP2
       instruments that measured refractory black carbon (BC), chlorine (Cl), sulfates ($SO_4$), organics
       (ORG), nitrates ($NO_3$), and ammonium ($NH_4$). Specifically, a MBL pass over the Pacific Ocean
       east of the Philippines was used to characterize the MBL aerosol chemistry in the absence of

anthropogenic aerosol (Fig. 5A). A pass directly within Manila's boundary layer was used to
       characterize recently emitted industrial and automobile emissions (Fig. 5B). The nearest pass
       through a BB plume over the Sulu Sea was used to characterize BB aerosol (Fig. 5C). Finally, a
       pass directly through the emissions plume of the R/V Sally Ride, which was conducting a
       complimentary project, the Propagation of Intra-Seasonal Tropical Oscillation (PISTON), was

used to characterize ship emissions (Fig. 5D).

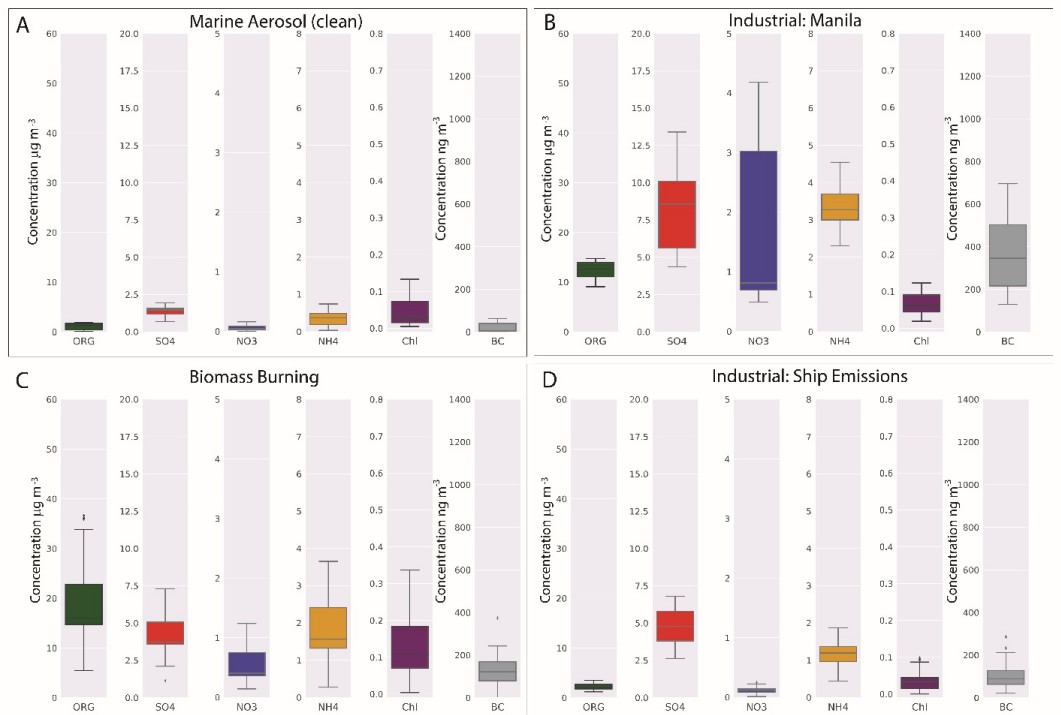

**Figure 5:** Chemical mass signatures of aerosol from boundary layer passes through (A) clean marine environment over the West Pacific Ocean east of the Philippines (01:25:28 - 01:35:28 UTC October 2019), (B) Manila industrial region (00:31:21 - 00:41:21 UTC 04 October 2019), (C) a biomass burning plume over the Sulu Sea (00:47:10 - 00:57:10 UTC 16 October 2019), (D) a pass through the *R/V Sally Ride* ship plume northeast of Luzon (04:03:42 - 04:13:42 UTC 2 October 2019).

The clean MBL had low concentrations of all chemical species. The Manila boundary layer chemical composition was dominated by higher concentrations of $NO_3$, $SO_4$, $NH_4$, and BC with median values of $NO_3$ (0.9 µg/m$^3$), $SO_4$ (8.7 µg/m$^3$), $NH_4$ (3.3 µg/m$^3$), and BC (381.3 ng/m$^3$). The most prominent feature is the large presence of $NO_3$, which likely formed from automobile combustion. The elevated values of ORG and BC are most likely from diesel exhaust and local BB (Bond et al., 2004; Kecorius et al., 2017). Although ORG and BC were elevated in the Manila boundary layer, it is unlikely that they would be from large BB events since none were influencing that region during the time of sampling based on backward HYSPLIT trajectories. The key



chemical signature of BB is elevated ORG. The median values of ORG and all other species within the BB plume were ORG (17.2 µg/m$^3$), NO$_3$ (0.5 µg/m$^3$), SO$_4$ (3.7 µg/m$^3$), NH$_4$ (1.6 µg/m$^3$), and BC (163 ng/m$^3$). Shipping emission, in the absence of other sources, displayed elevated concentrations of SO$_4$ and NH$_4$ with median values of SO$_4$ (4.9 µg/m$^3$) and NH$_4$ (1.2 µg/m$^3$), and

minimal concentrations of ORG, NO$_3$, and BC. These characteristics were used to categorize the remaining MBL passes.

Additionally, the three aerosol regimes were defined by categorizing and analyzing the 112 MBL passes from 19 research flights through a scikit-learn K-Means cluster algorithm, taking into consideration all six chemical signatures. The K-Means cluster centers identified four different

groupings. Each individual cluster's chemical composition was apparently dominated by either marine (clean), Manila industrial, BB, or ship emissions/aged industrial aerosol. In all cases, marine aerosol were present as the research flights were all over the ocean, but in the cases of Manila industrial, BB, and ship emissions, elevated levels of specific aerosol chemical species were found. To provide additional evidence that the suspected source regions were consistent with

the chemical signatures and groupings, 100-hour HYSPLIT backward trajectories were run for each MBL pass to confirm that the hypothesized primary aerosol source region was consistent with the interpretation of the chemical signatures (Fig. 6). Clean marine MBL passes (Fig. 6A) mostly originated 100 hours earlier from air mass sources over the West Pacific ocean east of the Philippines. Three backward trajectories originated west of Borneo prior to the primary BB season

in Borneo and Indonesia and were not near any ships or shipping lanes. Shipping emissions were from MBL passes over known shipping lanes (see methods in the appendix and Fig. 4) from the R/V Sally Ride (Fig. 6B). These may have been combined with aged industrial aerosol from mainland Asia over the South China Sea, but as will be shown, the NO$_3$ signature of automobile emissions found near Manila were not present over the South China Sea, suggesting that most of

the aerosols sampled came from ship emissions or aged industrial aerosol from which NO$_3$ had decayed. This region also and had remnants of elevated SO$_4$ from aged and secondary aerosol formation (Crosbie et al., 2022). HYSPLIT backward trajectories associated with industrial sources (Fig. 6B) were associated with air masses of origin from mainland Asia. Figure 6C confirmed that the BB MBL legs were sampled during a large BB event on 15 -16 September 2019

that occurred throughout Indonesia, Brunei, and Malaysia (Fig. 1).



## 4 Results

A total of 112 10-minute MBL passes were analyzed and categorized into the four categories, 49 marine, 10 BB, 7 passes in the Manila plume, and 46 passes sampling shipping emissions/aged and industrial aerosol (Table 1). Unfortunately no cloud base passes were made near Manila or within the Manila plume. Figure 5B is representative of the chemical signatures of the Manila plume for the seven passes through the plume in the boundary layer. These were all conducted on the same research flight. The data for the remaining groups were consolidated with statistical summaries presented in Figure 7.

*Marine:* The clean MBL had minimal concentration of aerosol of all chemical species (Fig. 7A). Most of these passes were located over the open ocean away from major industrial or BB locations and shipping lanes (Fig. 2). Marine passes were sampled away from major active BB sources and industrial centers, as confirmed by both HYSPLIT and the ERA5 winds. There were 49 clean MBL passes. The compositional chemistry of clean MBL sampling had median values of 2.2 $\mu g/m^3$ of ORG, 2.3 $\mu g/m^3$ of $SO_4$, 0.1 $\mu g/m^3$ of $NO_3$, 0.3 $\mu g/m^3$ of $NH_4$, 0.04 $\mu g/m^3$ of Cl, and 7.4 $ng/m^3$ of BC (Fig. 7A).

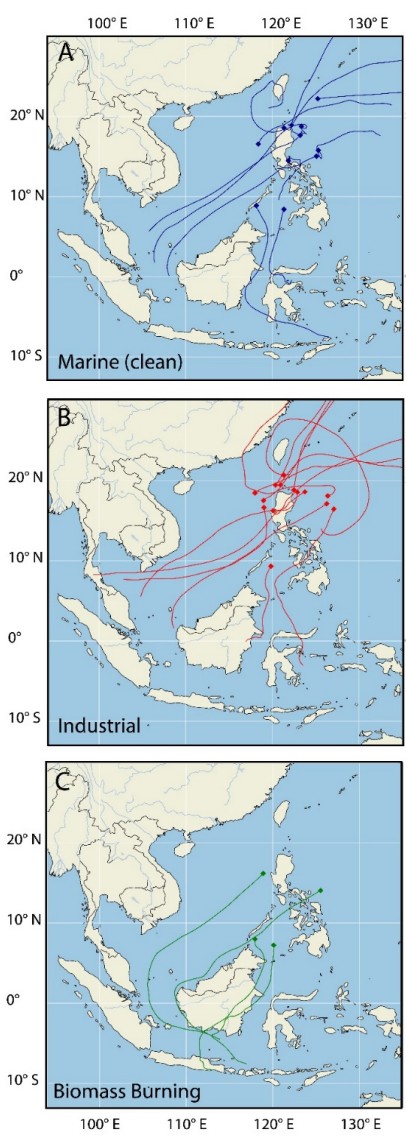

**Figure 6:** Example ensemble-averaged NOAA HYSPLIT 100 hour backward trajectories from 19 research flights between 24 August 2019 and 5 October 2019 within the marine boundary layer categorized by aerosol regime. A: Marine (Clean), B: Industrial, C: Biomass Burning.





**Table 1:** The total number of 10-minute boundary layer passes, and number of seconds sampling in cloud just above cloud base, categorized into their corresponding aerosol source type from all 19 research flights.

| Aerosol Source Type | # of 10 minute MBL passes | Number of seconds in cloud at elevations between 466 – 700 m |
| --- | --- | --- |
| Marine (Clean) | 49 | 747 |
| Biomass Burning | 10 | 401 |
| Industrial | 46 | 268 |
| Manila | 7 | 0 |
| **Total** | **112** | **1416** |


*Ship emissions/aged industrial aerosol::* Ship emission aerosol were identified when the P-3 flight path intersected ship plume projections or near-ship locations over the ocean (see supplemental video). When the P-3 was in the MBL within 60 km of a cargo or tanker ship and intersected its projected ship plume or sampled directly over shipping lanes, the MBL pass and cloud base pass were recorded as influenced by ship emissions (Fig. 4). A majority of the ship emission aerosol were sampled between 30 min – 4 hrs after being emitted from the ships. The ship emissions were likely mixed with aged aerosol from sources over Southeast Asia. Industrial anthropogenic aged aerosol away from shipping lanes were also sampled over the East China Sea. These over ocean samples likely originated from mainland Asia and Taiwan based on HYSPLIT backward trajectories. These were nearly all sampled late in the project after the retreat of the southwest monsoon over the Philippine region. The aerosol chemical composition influenced by ships and distant industrial sources had lower concentrations of $SO_4$, $NH_4$, and particularly $NO_3$ compared to aerosol measured near metro Manila (Fig. 5B). There were 46 legs associated with ship emissions/aged industrial pollution aerosol. The compositional chemistry of these aerosol had median values of 2.3 µg/m$^3$ of ORG, 6.1 µg/m$^3$ of $SO_4$, 0.1 µg/m$^3$ of $NO_3$, 1.4 µg/m$^3$ of $NH_4$, 0.04 µg/m$^3$ of Cl, and 74.2 ng/m$^3$ of BC. Dominant species in these industrial MBL legs were $SO_4$ and $NH_4$ (Fig. 7B).

*Biomass Burning:* BB passes all showed high concentrations of ORG and BC aerosol. The passes observing BB aerosol were over the Sulu Sea during a prominent BB event in Borneo on 15



September 2019 and just east of southern Luzon and Samar Islands on 16 September 2019. HYSPLIT backward trajectories and ERA5 reanalysis both indicate the aerosol in MBL were from the Borneo region. Given the backward trajectories and the high concentrations of BC, these MBL passes were indicative of BB aerosol (Bond et al., 2004; Massoli et al., 2015; Crosbie et al., 2022). The compositional chemistry sampled from the 10 BB MBL legs had median values of 21.2 $\mu g/m^3$

of ORG, 4.9 $\mu g/m^3$ of $SO_4$, 0.5 $\mu g/m^3$ of $NO_3$, 2.1 $\mu g/m^3$ of $NH_4$, 0.14 $\mu g/m^3$ of Cl, and 135.1 $ng/m^3$ of BC (Fig. 7C).



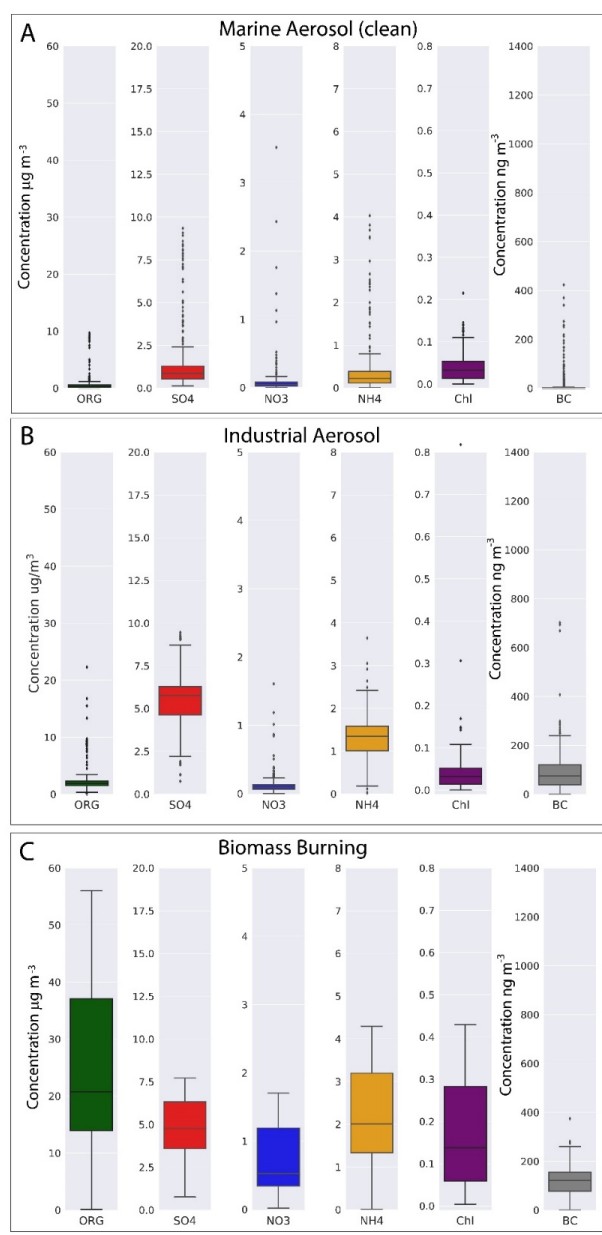

**Figure 7:** Averaged compositional chemistry samples from the aerosol mass spectrometer and single soot photometer depicting the difference in aerosol chemistry regimes. A: Clean marine, B: Ship emissions/aged industrial, C: Biomass Burning.



## 5 Cloud Base Measurements

### 5.1 Cloud Sampling and Statistics

Cloud sampling during CAMP[2]Ex was conducted in small warm cumulus and congestus clouds. The horizontal transects of clouds during cloud base passes ranged from 0.1 km to 4.5 km with most clouds in the range of 0.2 - 0.3 km (Fig. 8A, B). The High Spectral Resolution Lidar (HSRL) (Sawamura et al., 2017; Burton et al., 2018) and the Research Scanning Polarimeter (RSP; Cairns et al., 1999) showed that 50% of all transect lengths at all altitudes were < 0.6 km in length.

A cloud base pass with the FCDP was recorded if the cloud number droplet concentration ($N_c$) > $10\,cm^{-3}$ and liquid water content (LWC) was > $0.05\,g\,m^{-3}$. In-situ measurements from the FCDP showed that 50% of the cloud base transect lengths to be < 0.2 km, and 95% < 1.0 km (Fig. 8).

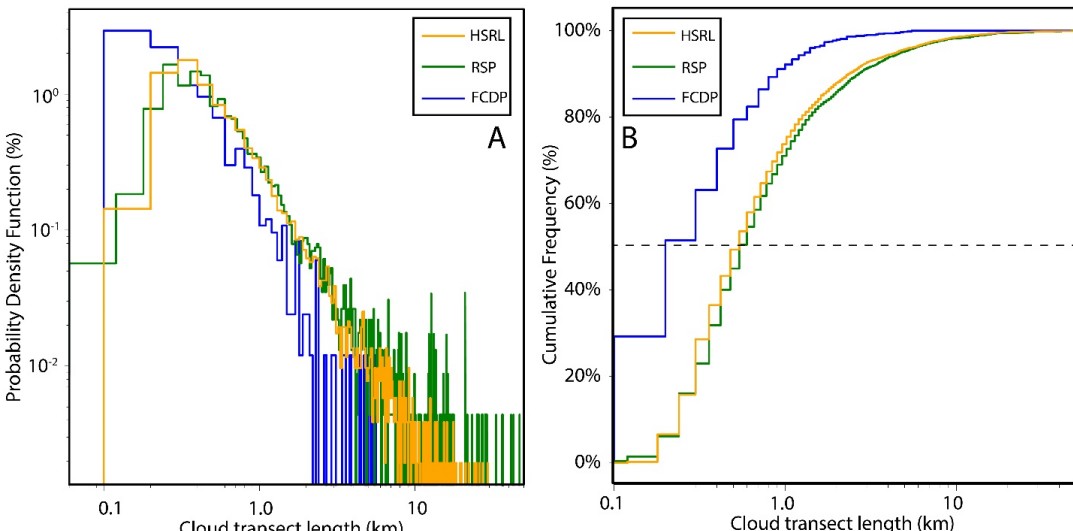

**Figure 8:** (A) The distribution of all cloud transects sampled on all 19 research flights by two remote sensing instruments, the High Spectral Resolution Lidar (HSRL, orange) and the Research Scanning Polarimeter (RSP, green). The Fast Cloud Droplet Probe (FCDP, blue) transect lengths are only for the passes just above cloud base; (B) cumulative frequency diagram of the cloud transect lengths.

When sampling near cloud base, the stage in the lifetime of the sampled cloud was unknown. It was also not possible to correctly identify what part of the cloud was sampled, whether





the edge or the core updraft. Updraft strengths just above cloud base, measured by TAMMS ranged from 0.1 to 3.0 ms$^{-1}$ (Fig. 9A). Median updraft speeds did not differ greatly in the clouds sampled over the three oceanic regions, the West Pacific (0.4 ms$^{-1}$), the Sulu Sea (0.4 ms$^{-1}$), and
the South China Sea (0.5 ms$^{-1}$). Based on figure 9B, 50% of the updrafts sampled had vertical velocities exceeding 0.4 ms$^{-1}$. To ensure that clouds sampled were drawing air from the MBL and were near the core of the updraft, only cloud base passes with updrafts > 0.4 ms$^{-1}$ were included in the subsequent analysis.

Nearly all cloud base passes were completed in the same region immediately following
MBL passes (Fig. 2). There were two legs with cloud base passes over the southern Sulu Sea where the cloud base passes were delayed to sample growing clouds to the north. The aircraft then returned to the location of the MBL passes and sampled the cloud base.

In total, 1416 seconds of cloud base passes were categorized into the three aerosol regimes (Table 1). There were no clouds sampled at cloud base during flights around the city of Manila, so
all cloud base passes categorized as ship emissions/aged industrial were sampled over the open ocean.

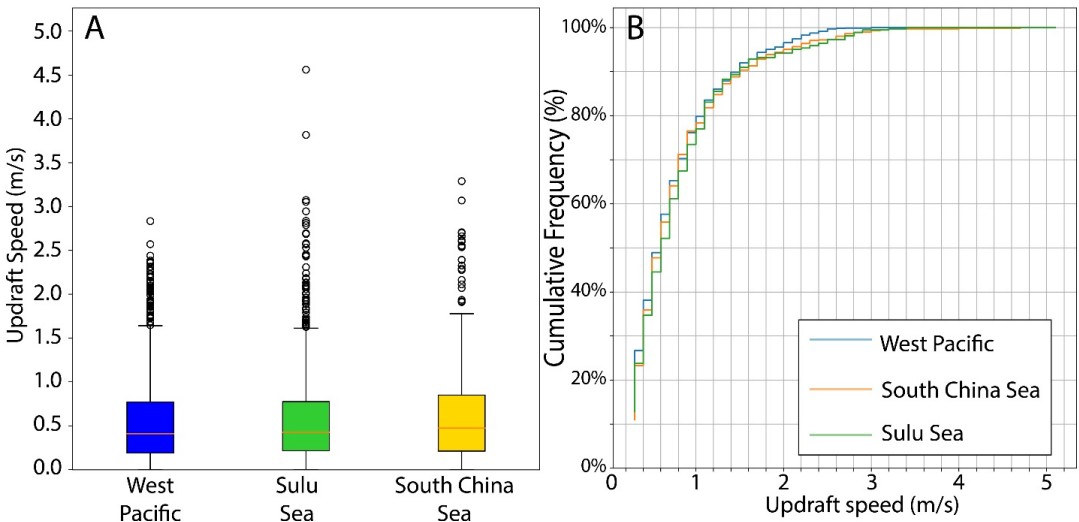

**Figure 9:** (A) The distribution of the updraft speeds measured at cloud base using data from the Turbulent Air Motion Measurement System (TAMMS) over three oceanic regions around the
Philippines. The median (red line), the 25[th] and 75[th] percentile (colored boxes) the 5[th] and 95[th]





percentile (black whiskers), and outliers (circles) are shown. (B) Cumulative frequency diagram of updrafts over the three regions.

## 5.2 Cloud droplet size distributions

For clouds with updrafts > 0.4 ms$^{-1}$, the median cloud droplet number concentration just above cloud base for the clean marine clouds was 36.3 cm$^{-3}$, while industrial was 112.2 cm$^{-3}$, and BB was 251.2 cm$^{-3}$ (Fig. 10). The 75$^{th}$ percentile values for marine clouds was 63.5 cm$^{-3}$, the industrial 273.8 cm$^{-3}$, and BB was 541.1 cm$^{-3}$, while the 95$^{th}$ percentile values for marine clouds was 149.4 cm$^{-3}$, industrial 788.0 cm$^{-3}$, and BB was 1308.7 cm$^{-3}$.






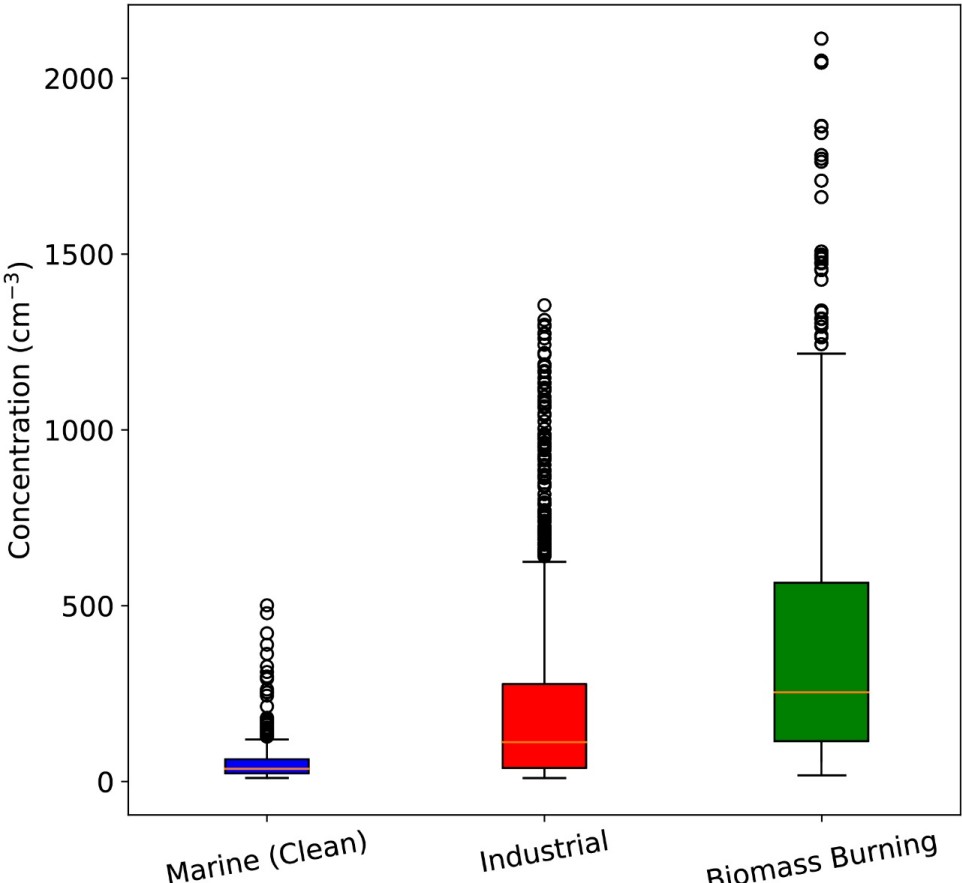

**Figure 10:** Cloud droplet concentration just above cloud base for the three aerosol source regions. The median (orange line), the $25^{th}$ and $75^{th}$ percentile (colored boxes) the $5^{th}$ and $95^{th}$ percentile (black whiskers), and outliers (circles) are shown.

The cloud droplet size distributions for the marine category, normalized by LWC are shown in Figure 11A. The remaining panels (Figs. 11 B-C) show the normalized cloud droplet size distributions for the other two aerosol source regions together with the marine spectra. Clouds impacted by BB and ship emission/aged industrial aerosol contained higher concentrations of



cloud droplets in size ranges corresponding to bins < 13 µm, with concentrations in the 8.0 - 10.0 µm size bin almost 1.5 orders of magnitude greater than the marine clouds. At size ranges between 13.0 - 34.5 µm the concentrations in all categories were almost an order of magnitude less than the

marine category. In the size bins > 34.5 µm concentrations were equal to, or slightly exceeded, the concentrations of the marine clouds. In the largest bins no drops were recorded for the BB category, while the ship emission/aged industrial aerosol category had a higher concentration of droplets in all three of the largest bins. Cloud droplet spectra influenced by ship emissions had a broadening of droplet distribution (Fig. 11C), which is similar to Ackerman et al., (2000b) findings

for ship tracks when compared to ambient maritime values.



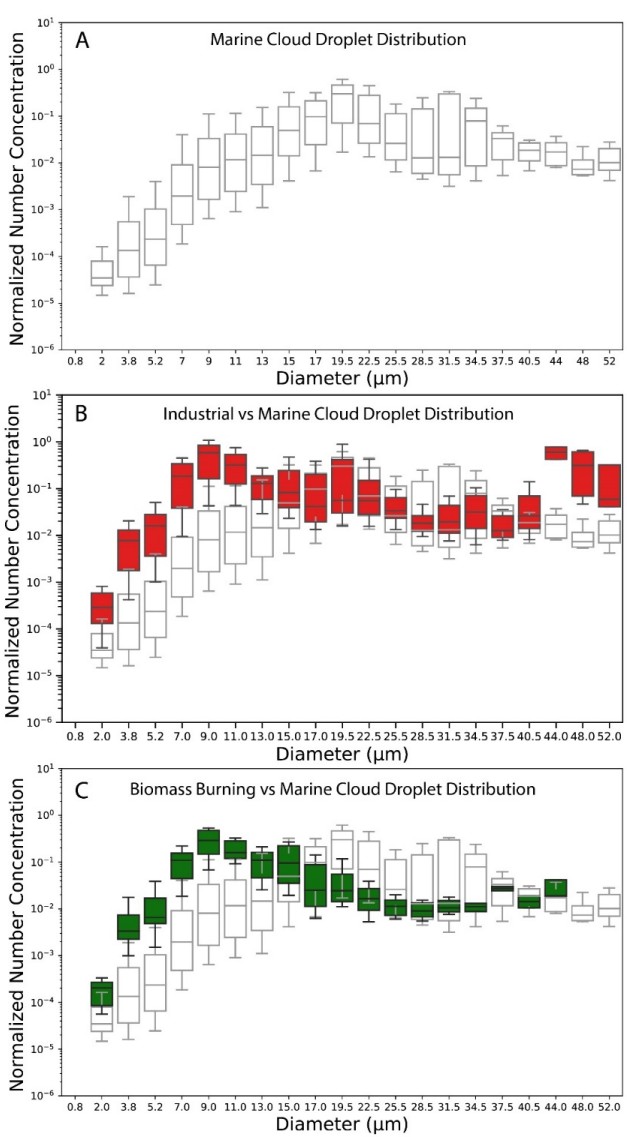

**Figure 11:** Mean cloud number concentration spectra normalized by liquid water content from the fast cloud droplet probe (FCDP) just above cloud base for the three aerosol categories. (A) Cloud base spectra in clean marine environments (white) compared with (B) industrial (red), (C) biomass burning (green).

The marine cloud droplet distribution, as well as the other distributions, most likely contained large sea salt aerosol as CCN, producing droplets with diameters > 40.5 µm. Crosbie et



al., (2022) noted that sea salt was a dominant mass component in in-situ collected cloud water during CAMP$^2$Ex flights. A few of the marine clouds sampled just above cloud base contained

droplet concentrations $> 150$ cm$^{-3}$ (see outliers in Figure 10). These were most likely clouds that were impacted somewhat by aerosol sources of uncertain origin.

## 6 Conclusions

The Clouds, Aerosol and Monsoon Processes – Philippines Experiment (CAMP$^2$Ex) provided an opportunity to examine the impact of different aerosol sources on cloud base microphysical

properties using data from 19 research flights flown around the Philippine islands between 24 August – 5 October 2019. In total 112, 10-minute marine boundary layer (MBL) legs from the aerosol mass spectrometer were analyzed. Four different aerosol source regions were identified within the MBL from chemical analysis of the aerosol, HYSPLIT backward trajectories, cargo and tanker ship emission projections, and a K-mean cluster algorithm. The Manila aerosol source

region did not have cloud passes to determine the impact of the Manila aerosol plume on cloud base microstructure. Cloud droplet size distributions from the remaining three regions influenced by different aerosol sources were measured using the Fast Cloud Droplet Probe (FCDP). Small cumulus and congestus clouds were analyzed for this study. Over 50% of the cloud transects were $< 0.2$ km, with 95% $< 0.9$ km in length. The cloud droplet spectra analyses was restricted to those

periods where the measured updraft was $> 0.4$ ms$^{-1}$ to ensure that the core updraft was sampled and the cloud was drawing air upward from the MBL. There were 1416 seconds of cloud base sampling meeting this criterion. The key findings of this analysis are:

1. Four sources were found to influence the aerosol chemical composition in the Philippine region. These were marine (ocean source), ship emissions mixed with aged industrial

aerosol from urban sources in mainland Southeast Asia, fresh industrial and automobile-generated aerosol from the city of Manila, and aerosol from biomass burning originating from Borneo and Indonesia.

2. The marine aerosol chemical composition, which was not influenced by anthropogenic aerosol, had low values of all sampled chemical signatures, specifically median values of

$2.2\,\mu g/m^3$ of organics (ORG), $2.3\,\mu g/m^3$ of $SO_4$, $0.1\,\mu g/m^3$ of $NO_3$, $0.3\,\mu g/m^3$ of $NH_4$, $0.04$ $\mu g/m^3$ of Cl, and $7.4$ ng/m$^3$ of refractory black carbon (BC).



3. The key chemical signatures of the other three aerosol source regions were (1) ship emissions/aged industrial: elevated $SO_4$ concentrations with a median value of 6.1 µg/m$^3$; (2) biomass burning: elevated concentrations of ORG 21.2 µg/m$^3$ and BC 135.1 ng/m$^3$. (3) Manila: median values of 0.9 µg/m$^3$ of $NO_3$.

4. Normalized cloud droplet size distributions showed that clouds impacted by ship emissions and aged industrial aerosol, and biomass burning contained higher concentrations of cloud droplets by as much as 1.5 orders of magnitude in the size ranges < 13 µm compared to marine clouds.

5. At size ranges between 13.0 - 34.5 µm the median concentrations in all categories were nearly an order of magnitude less than the marine category. For droplets with diameters > 34.5 µm, concentrations were equal to, or slightly exceeded, the concentrations of the marine clouds.

These analyses show that the anthropogenic aerosol generated from industrial, ships, and biomass burning sources have a significant influence on the cloud base microphysical structure of clouds in the Philippine region, particularly over the South China Sea. Future studies will examine how these changes in cloud droplet spectra as a result of aerosol pollution manifest in the higher regions of the clouds and impact precipitation, radiative properties, and lifetime in small cumulus and congestus clouds.

*Data Availability.* All CAMP$^2$Ex in situ data used in this study are publicly available at https://www-air.larc.nasa.gov/cgi-bin/ArcView/camp2ex
DOI:10.5067/Suborbital/CAMP2EX2018/DATA001.
The ERA5 data (https://doi.org/10.5065/BH6N-5N20: 02 April 2022) are downloadable. Cargo and tanker ship data can be ordered from http://www.astrapaging.com/.

*Author Contributions.* RMM, RMR, and LDG conceived the study design and analysis. RMM analyzed the data with inputs from RMR, LDG, GMM, and SWN. RMM, GMM, and LDG acquired funding. LZ, SW, and KT collected data on board the NASA P-3. DF provided HSRL and RSP derived data. MR analyzed ship plume projections. RMM wrote the paper with reviews from co-authors.

*Acknowledgements.* The authors wish to acknowledge the entire CAMP$^2$Ex science team, NASA Ames Earth Science Project Office and the NASA P-3 crew for the successful deployment.



In addition, we would like to thank Michael Shook for creating the merged instrument data files for the CAMP²Ex campaign. We would also like to thank the NASA Ames Earth Science Project Office for their endless help and support throughout the mission.

*Financial support.* Funding for this project was from NASA Award 80NSSC18K0150. RMM was supported by NASA headquarters under the NASA Future Investigators in NASA Earth and Space Sciences and Technology grant 80NSSC19K1371, 80NSSC18K0144, and 80NSSC21K1449.


## Appendix

This appendix describes the methods used to produce a dataset with the predicted locations of cargo and tanker ship aerosol in the vicinity of the P-3 aircraft during CAMP²Ex. The dataset provides information on the P-3 MBL status, the distance from Manila, the number

of ships within a 60 km and 100 km radius of the P-3, the number of discrete plumes within 60 km and 100 km of the P-3, the time of a plume-aircraft intersection (if such an intersection occurred), the age of the intersected plume, and the MMSI location of the ship that produced that plume.

*Cargo and Tanker Ship Data:* The MMSI ship dataset purchased from Astra Paging Ltd provided

ship information covering the region of flights around the Philippines at 3 hr frequency between the hours of 22:00:00 UTC and 9:00:00 UTC the next day. The heading, course, and speed information of each ship was used to estimate coordinates of the ships at 1 Hz resolution using the World Geodetic System 1984 Coordinate Reference System (WGS84 CRS), with precautions made for ships that were projected to arrive on land.

*Initial backwards projection:* To initiate the ship position, ship locations before 22:00:00 UTC were projected backwards in time to the hour of 15:00:00 UTC to prepare complete predicted plume positions present before the P-3 takeoff. Using the earliest reported position of each ship (referred to here as a ping), all ships were projected to predicted locations at 15:00:00 UTC. To do this, each course was flipped by 180°, i.e., in the opposite direction. Ship speed and the duration



of time between 15:00:00 UTC and the earliest position at 22:00:00 UTC was used to produce a travel distance. Paired with the flipped course, a geodesic was used to project each ship's coordinates using the WGS84 model. If a ship was projected to appear on land, it was ignored. Only two ships were projected to arrive on land across the dataset. For these two ships, they were placed at the coordinates reported by its earliest location, and were given a speed of 0 kts between

15:00:00 and 22:00:00 UTC. Thus, the ship was treated as stationary until time progressed to its earliest report, at which point the ship was given its reported speed (Fig. 4).

*Projection of ship positions to 22:00:00 UTC:* After this initial back projection, ships positions were projected via the geodesic generated from their course and speed. The time duration between each update was one second.

*Treatment of ship pings:* Once the hour of 22:00:00 UTC was reached, ship ping data was used to update the ships' position. At each time step, the ship dataset was checked for a ping. Ship projections were overwritten with the relevant information given by the ping – the latitude, longitude, course, and speed. The ping data for each ship is not continuous, there were some cases where the coordinates described by a ship ping were much further away from the ship's previous

coordinates, far enough that it would be impossible for the ship to traverse this distance during the timestep. Ships that move with a velocity greater than 50 kts (approx. 25 ms$^{-1}$) were labelled as "teleporting". This phenomenon was taken into account when generating plumes, specifically plume lines.

*Ship Plumes:* The aerosol plumes produced by each ship were treated discretely. Every 600

seconds, a plume was generated at the location of each ship. Wind data from the ERA5 level 1 reanalysis was used to calculate plume advection. Much in the same way that the ship positions were projected, a geodesic was used to determine a plume's expected coordinates. The u and v wind components were used to find the azimuth and length of the geodesic used to project each plume. The time resolution of the ERA5 dataset is 1 hr. Since ships positions were estimated at 1

Hz, the step's current time was rounded to the nearest hour for indexing the ERA5 winds. The coordinate resolution of the ERA5 dataset is 0.25°, thus the indexable coordinate nearest to each plume was used for plume projection. Each plume was assigned an age, starting at zero when the plume is initially generated. Each second, the age was incremented. At an age of 14,400 sec (4





hrs), the plumes were assumed to have completely mixed out into the environment, and the plume
was terminated.

To approximate continuous aerosol production, lines were drawn between plumes based
on which ship produced them, producing a chain of plumes from each ship. Each link in the chain
was given an age equal to that of the younger plume to which it is connected.

As mentioned earlier, the intermittent shipping data causes some ships to appear to move
quicker than is possible. In the event that a plume was produced by a ship that had just "teleported",
the plume line that would connect this newly-produced plume with the previous plume is
discarded, creating a discontinuity in the ship's plume streak.

*Data Collection:* When the P-3 data was integrated with the ship plume projections, the following
information was recorded at each second of the research flight:

Boundary layer indication. This is simple boolean value representing if the P-3 aircraft was below
the MBL median altitude, 466 m. A value of 1 indicates that it was under 466 m; a value of 0
indicates that it was above.

Distance from Manila. This is the distance between the P-3 aircraft's present coordinates and
Manila, defined at (14.5995° N, 120.9842° E).

Ships within 60 km and 100 km. The number of ships within radii of 60 km and 100 km of the
aircraft were recorded.

Plume line intersection data. The path that the P-3 aircraft takes during the interval 30 sec before
and 30 sec after the present timestep was used to determine if the P-3 aircraft intersected a plume
line. If this path did indeed intersect a plume line, the time of this intersection, the age of the plume
line, and the MMSI of the ship that produced the plume were all recorded.

*Competing interests.* The contact author has declared that neither they nor their co-authors have
any competing interests.




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
