# Peer review of "Influence of natural and anthropogenic aerosol on cloud base droplet size distributions in clouds over the South China Sea and Western Pacific"

_EGUsphere, 2022_

## Referee Comment (RC2)

L29.  Add s to show.  Insert that before three.
L31.  Insert clean before marine.
L36-7.  Move based on chemical signatures to the beginning of the sentence.  Comma after emissions.  Change The ship emissions to which.
L41.  Delete sizes with.  Insert clean before marine.
L77, 260 & 386.  Three different symbols for droplet concentrations.  Use only the last one, $N_c$.  $N_d$ should be reserved for drizzle drop concentrations, which you say will be the subject of your next paper.
L87.  Delete on a.  Delete basis.  Add ly to constant.
L88.  Add Durkee et al. (2000).
L98.  Add Hudson et al. (2000).
L115.  Aircrafts singular.
L116-119.  Discussion of the Learjet is not needed.
L133.  Add comma after fires.
L207.  Remove also.
L223.  Change in to is.
L228 and other lines.  At least somewhere denote local time at least relative to UTC.
L253.  Change daytime to daylight.  Note local time.
L258-9.  Move at 1 Hz resolution (Fig. 3B) before above.  Insert and right before above.
Figure 5.  I can see the median lines in some but not all.  Not in BB ORG & SO4 and Industrial SO4.
L302.  Insert done after all.
L316.  Delete and.
L318.  Add n to Asia and then move Asian in front of origin followed immediately by mainland.  Delete from.
L329.  Concentration plural.
Figure 7.  Can't see median line in BB NO3.
L428.  Explain normalized.
Figure 11.  Explain normalized.  Is the ordinate dN/dlogD?  Or what is it?  At least it must be per $cm^3$
L448.  Change in to of.
L461. Change microstructure to microphysics.
Appendix is difficult to follow.
L524.  Define MMSI.
L535-6.  Duration and of time are redundant.
L537.  Is this going into port?
L540.  Time progressed is redundant.
L541.  Explain given its reported speed.
L545.  Was reached seems unnecessary.  Just say At 22:00 UTC.
L554.  Change discretely to separately.  The former sounds like a secret.
L556.  In could be deleted.
L557.  Explain geodesic.

Durkee, P.A., K.J. Noone, R.J. Ferek, D.W. Johnson, J.P. Taylor, T.J. Garrett, P.V. Hobbs, J.G. Hudson, C.S. Bretherton, G. Innis, G.M. Frick, W.A. Hoppel, C.D. O'Dowd, L.M. Russell, R. Gasparovic, K.E. Nielsen, S.A. Tessmer, E. Öström, S.R. Osborne, R.C. Flagan, J.H. Seinfeld, and H. Rand, 2000: The Impact of Ship-Produced Aerosols on the Microstructure and Albedo of Warm Marine Stratocumulus Clouds: A Test of MAST Hypotheses 1i and 1ii. *J. Atmos. Sci.,* **57**, 16, 2554–2569 DOI: 10.1175/1520-0469(2000)057<2554:TIOSPA>2.0.CO;2

Hudson, J.G., T.J. Garrett, P.V. Hobbs, S.R. Strader, Y.X. Xie, and S.S. Yum, 2000: Cloud condensation nuclei and ship track clouds. *J. Atmos. Sci.*, **57**, 2696-2706.

---

## Author Comment (AC1)

Response to the reviews on "Influence of natural and anthropogenic aerosol on cloud base droplet size distributions in clouds over the South China Sea and Western Pacific"

We would like to thank both reviewers for providing comments and suggestions to this manuscript. Reviewer comments are in black text, responses to comments is in red text, and added text to the manuscript is in blue text.

**Reviewer 1 comments:**

General comments:

This paper discussed the different aerosol sources' influence on cloud base droplet size distribution using a comprehensive dataset from P3 airplane. The data is extremely valuable for improving the current understanding of the aerosol-cloud interaction, especially over the South China sea and the western pacific. The paper is well written and made an excellent measurement report. However, the data analysis did not use the full potential of such a rich dataset nor provide new insight into the aerosol-cloud interaction. Thus, if authors are willing to explore the linkage between the meteorological and climate features in the southeast Asia region and the observation. This paper can contribute more as a research article. Otherwise, it is in good shape as a measurement report after addressing the comments below.

We added a paragraph describing the meteorology over the sampling region during the campaign. We also added a reference to Reid et al., 2023, a comprehensive summary of the CAMP$^2$Ex project which includes a discussion of the meteorology during CAMP$^2$Ex. The winds during CAMP$^2$Ex did not vary much day to day in each of the two monsoon periods, so we did not think it was necessary to describe the meteorology for each day during the project. The new paragraphs are given below:

"The first part of CAMP2Ex (Aug 24 - Sept 22, 2019) occurred when the southeast Asian monsoon trough was located north of the Philippines, and the flow was directed from the maritime continent northeastward, transporting aerosol associated with biomass burning toward the sampling region. The second part of CAMP2Ex (Sept 23 - Oct 5, 2019) occurred after the monsoon trough retreated south of the archipelago and northwest monsoon flow moved across the sampling region (see figure 2 in Reid et al., 2023). The monsoon flow during CAMP2Ex was impacted occasionally by the passage of tropical cyclones which moved westward north of Luzon along the monsoon trough.

During both monsoon periods, clouds over the sampling regions were primarily small maritime cumulus, with deeper convection, and associated cold pools developing, for example, by heating over local islands or lifting along cold pool outflows."

Specific comments:

Abstract: "three primary sources influenced the aerosol chemical composition: marine (ocean source), industrial (Southeast Asia, Manila, and cargo and tanker ship emissions), and biomass burning (Borneo and Indonesia)." The industrial is misleading. The paper discussed ship emissions.

Our definition of industrial appeared in the abstract of the original paper on line 30 and included aerosol from Southeast Asia, Manila, and cargo and tanker shipping emissions. The chemical signatures of industrial emissions from Manila were shown on Figure 5B in the original paper, with a discussion of the Manila aerosol chemistry on original lines 283-290. The long range transport of anthropogenic aerosol from south east Asia was discussed on lines 35,135, 151, 311-319, 351-355, 468-472 of the original paper. These discussions are now located on lines XX (add after done). We believe that we were clear on what we meant by industrial.

Introduction: The current result is consistent with the previous studies the author quoted in the introduction. "The impact of anthropogenic aerosols such as sulfate, nitrate, and BC has been a main topic of interest for many years as they lead to an increase of CCN that increases the cloud droplet number concentration (Nd) and decreases the effective radius (re) of the droplets, producing more reflective clouds for the same liquid water path.

For example, Radke et al., (1989) observed an increase in total cloud droplet concentrations, but a decrease in cloud droplet sizes in clouds over shipping lanes. Cloud droplet number has also been reported to increase with aerosol loading over the East China Sea (Bennartz et al., 2011)." It is more scientifically significant if the authors can further explore the current results.

We used all the available data from the aircraft and shipping data in the paper. We have added information about the meteorology of the region. Our group is exploring other aspects of the data related to remote sensing, cloud dimensions and geometry, aerosol and cloud retrievals from satellites, radar, lidar. These will appear in other publications (e.g. Fu et al., 2022). We believe the work in this paper provides a clear scientific analysis of the impact of aerosol chemistry on cloud base properties. We don't believe the paper fits into the category of a measurement paper.

Page 6, lines 121-129. How are those meteorological and climate features related to the aerosol-cloud interaction? It will be very insightful to make the linkage.

We added the paragraph explaining the meteorology during the CAMP$^2$Ex project. The HYSPLIT trajectories in figure 6 show the change of the flows into the sampling area during the southeast and northwest monsoon periods

Page 8, section 2.1. How many flight hours or data points were collected in each category?

The information the review results is in Table 1 and is referenced at the first sentence of the results section (section 4).

What criteria do you use to characterize the influence of each aerosol source? Hysplit trajectory? Aerosol concentration? or chemical loading? Or combination?

We used the aerosol chemistry measurements below cloud base to characterize the aerosol types summarized on figure 7. We confirmed these with Hysplit trajectories to ensure our interpretations were consistent with source regions. This is described in results section 4.

Page 11, line 203. Please define "nearest", as one hour apart or 10 hours apart? Will that affect the distribution in figure 3?

The procedure we used to calculate the mean LCL was poorly described. We revised this section. The phrase the reviewer flagged is no longer there.

Page 13, section 2.5. How does this hysplit data link to three aerosol sources?

The link between the HYSPLIT trajectories and the aerosol source regions is described in the results section 4 beginning on page 19 line 329 and ending on page 22 line 371 in the original paper and is shown on figure 6 of the original paper.

Figure 4, line 246-251. It is hard to separate the thick and dashed lines in the figure. Maybe use a different color for the altitude < 466 or >466? Maybe also use a larger ring size.

In developing this figure we experimented with a number of different lines thicknesses and approaches to showing the ship tracks. This was the best version we came up with after quite a bit of experimentation. We also developed the animation in the supplement based on this best solution. Since we already explored this other options we decided to keep the figure as is.

Line 256. AMS has a 30-second resolution. Why do you exclude anything shorter than 10 mins?

Our choice of 10 minutes was based on recommendation from the Langley aerosol research group based on their experience using the aerosol mass spectrometer. We added the following sentence:

The instrument team recommended a minimum 10-minute averaging interval (20 data points) to obtain representative chemical signatures.

Line 261. How does the MBL passes related to the cloud base passes? At the same region? Altitude/longitude boundaries? Please clarify.

Line 404 on the original paper states the following: "Nearly all cloud base passes were completed in the same region immediately following MBL passes (Fig. 2). There were two legs with cloud base passes over the southern Sulu Sea where the cloud base passes were delayed to sample growing clouds to the north. The aircraft then returned to the location of the MBL passes and sampled the cloud base."

In addition, here used 1416 passes and later, 1416 seconds were used. Please correct. For 112 MBL passes, how long does each pass last?

The text was corrected and the information added. The sentence now reads:

"A total of 112 10-minute MBL passes and 1416 seconds of cloud base passes were recorded."

Line 305. I am confused about the procedure for determining the suspected source regions. Please clarify. How many backward trajectories do you run for each pass? Why choose 100 hours? How do you separate the long-term transport influence from regional influence with 100-hour trajectories?

We chose 100 hour trajectories to ensure that the trajectories reached aerosol source regions over land, either the maritime continent or East Asia. For short trajectories, the trajectories remain over the ocean and source regions cannot be identified. In some cases even after 100 hours the trajectories were still over the ocean and not near any land mass suggesting a marine source of aerosol. Aside from shipping emissions, and the Manila metro, there are no other regional influences except for small islands.

Table 1. Please correct. Number of passes or seconds? Please consider adding information about linking each MBL pass with cloud passes, such as the latitude range when sampling. For example, if MBL passes took 490 mins and their corresponding cloud passes only 747 seconds (about 12.5 mins). Does the result meaningfully capture the influence? Please explain.

Table 1 was correct. The text on line 261 was incorrect and was fixed.

Figs 5 and 7 provide similar info. Please explain the significant scientific contributions for including both or move one to a supplemental document.

Figure 5 shows individual 10-minute MBL pass data. Figure 7 shows the average of all the 10-minute MBL passes in each of the three categories.

Page 23, section 5.1. HSRL and RSP were discussed here but not included in the Methodology section like FCDP. Please provide more info about the operation.

We added a short section introducing the instruments and provide references to the instruments and a reference to Fu et al., 2022. In Fu et al., 2022 the procedure to determine cloud transect length is described in detail. The following section was added:

2.3 Remote Sensing Instrumentation

Cloud transect length was determined three ways, using the FCDP, the NASA High Spectral Resolution Lidar (HSRL-2, Burton et al., 2018; Sawamura et al., 2017), and the Research Scanning Polarimeter (RSP, Cairns et al., 1999). The use of the instruments in CAMP2Ex and the approach to determine cloud transect length is presented in detail in Fu et al., 2022.

Line 398-400. What is the median updraft for the different aerosol-influenced clouds? Please consider re-characterize the cloud category to link them with the aerosol category.

We added a figure showing the distribution of updraft speeds for the different aerosol influenced clouds, and a discussion of the figure.

**Reviewer 2 comments:**

L29. Add s to show. Insert that before three.

The typo was corrected.

L31. Insert clean before marine.

Clean added.

L36-7. Move based on chemical signatures to the beginning of the sentence. Comma after emissions. Change The ship emissions to which.

Change made.

L41. Delete sizes with. Insert clean before marine.

Change made.

L77, 260 & 386. Three different symbols for droplet concentrations. Use only the last one, Nc. Nd should be reserved for drizzle drop concentrations, which you say will be the subject of your next paper.

Change made.

L87. Delete on a. Delete basis. Add ly to constant.

Change made.

L88. Add Durkee et al. (2000).

Reference added.

L98. Add Hudson et al. (2000).

Reference added.

L115. Aircrafts singular.

Change made.

L116-119. Discussion of the Learjet is not needed.

Discussion deleted.

L133. Add comma after fires.

Comma added.

L207. Remove also.

Change made.

L223. Change in to is.

Change made.

L228 and other lines. At least somewhere denote local time at least relative to UTC.

Philippine local time noted in the paper.

L253. Change daytime to daylight. Note local time.

Change made and time added.

L258-9. Move at 1 Hz resolution (Fig. 3B) before above. Insert and right before above.

Change made.

Figure 5. I can see the median lines in some but not all. Not in BB ORG & SO4 and Industrial SO4.

Figure updated to show median lines.

L302. Insert done after all.

Change made.

L316. Delete and.

Change made.

L318. Add n to Asia and then move Asian in front of origin followed immediately by mainland. Delete from.

Change made.

L329. Concentration plural.

Change made.

Figure 7. Can't see median line in BB NO3.

Figure updated.

L428. Explain normalized.

A sentence was added explaining the normalization procedure.

Figure 11. Explain normalized. Is the ordinate dN/dlogD? Or what is it? At least it must be per cm3
The units were added to the figure. They are $cm^{-3}\mu m^{-1}$.

L448. Change in to of.

Change made.

L461. Change microstructure to microphysics.

Change made.

Appendix is difficult to follow.

It was unclear what part of the appendix the reviewer found difficult to follow. No changes made.

L524. Define MMSI.

MMSI was defined on line 222 of the original paper.

L535-6. Duration and of time are redundant.

Fixed.

L537. Is this going into port?

Changed to projected to going into port.

L540. Time progressed is redundant.

Fixed.

L541. Explain given its reported speed.

Fixed.

L545. Was reached seems unnecessary. Just say At 22:00 UTC.

Fixed.

L554. Change discretely to separately. The former sounds like a secret.

Change made.

L556. In could be deleted.
Change made.

L557. Explain geodesic.

The word geodesic means relating to or denoting the shortest possible line between two points on a sphere or other curved surface.

---

## Referee Report (RR1)

The authors have complied with my suggestions. But in so doing have created more questions that have indicated further improvements.

L465-6. This is not normalization. LWC within a size bin must be considerably less than LWC of the entire spectrum. Thus, concentrations in all bins are reduced by this "normalization." This reduction would be greater for the smaller size bins that would have less LWC due to smaller sizes to the third power. Multiplying by the ratio of LWC within each bin to the mean or median LWC of the spectrum would be a normalization. Or concentrations could be normalized according to the size widths of the bins? This needs explanation. Is whatever normalization that was or should be applied a common practice in cloud microphysics research?

L475-6. This assertion requires explanation. Calculate the broader spectra and demonstrate. Is this over the entire size range or over some part of the droplet size range. Moreover, ship emissions are included in Fig. 12B not 12C, which is biomass. But even so panel B includes industrial. So, it is a further assertion to single out ships from industrial when both are included in Fig. 12B.

L484-489. This paragraph is an out of context assertion. In order to stand it requires data backup and proper context.

L36-38. This is an assertion that was not demonstrated.

Explain the meaning of Fig. 3. There are 10 plots in A & B. Are they related one-by-one to each other or is this just a coincidence? How are the two panels related? Apparently, you want to show that the cloud measurements were above the LCL.

L287-291. How do the emissions of a research vessel compare to those of cargo and tanker vessels? I doubt that Sally Ride used Bunker Fuel. What fuel did it use?

L603. Four hours seems like a short time for a ship plume to disappear. Is there a reference to this fact?

Minor suggestions:

L30. Insert clean before marine.

L33 & 35. $SO_4$ should not be the same. Consistent with L350 & L513 $SO_4$ should be 2.3 $\mu g/m^3$ in L33. Also consistent with L350 & L513 ORG should be 2.2 $\mu g/m^3$ in L32 and $NH_4$ should be 0.3 $\mu g/m^3$. The others in L32-3 are consistent with L349 & L350 and L523-4.

L43 & 45. Insert clean before marine.

L46. Add d to influence.

L63-64. Insert Hallett et al. (1989).

L77. Insert Hudson et al. (2009) and Hudson & Noble (2014).

I appreciate $N_c$. But now $N_c$ can be employed in L80, L276, L402 and L455.

L81. Insert Hudson & Yum (2002).

L86-7. Move constantly in front of into.

L107. Add Twohy et al. (2001).

L290. Complementary.

L331. Change aerosols to particles.

L339. Insert clean before marine.

L401. Delete in length.

L588. Remove away.

Hallett, J., J.G. Hudson, and C.F. Rogers, 1989: Characterization of combustion aerosols for haze and cloud formation. *J. Aeros. Sci. and Technol.*, **10**, 70-83.

Hudson, J.G., and S. Noble, 2014: CCN and vertical velocity influences on droplet concentrations and supersaturations in clean and polluted stratus clouds. *J. Atmos. Sci.*, *71*, 312-331. DOI: 10.1175/JAS-D-13-086.1

Hudson, J.G., S. Noble, V. Jha, and S. Mishra, 2009: Correlations of small cumuli droplet and drizzle drop concentrations with cloud condensation nuclei concentrations. *J. Geophys. Res.*, **114**, D05201, doi:10.1029/2008JD010581.

Hudson, J.G., and S.S. Yum, 2002: Cloud condensation nuclei spectra and polluted and clean clouds over the Indian Ocean. *J. Geophys. Res.*, **107(D19), 8022,** doi:10.1029/2001JD000829.

Twohy, C.H., J.G. Hudson, S.S. Yum, J.R. Anderson, S.K. Durlak, and D. Baumgardner, 2001: Characteristics of cloud nucleating aerosols in the Indian Ocean region. *J. Geophys. Res.,* **106, D22***,* 28699- 28710**.**

---

## Author Response (AR2)

Response to the reviews on "Influence of natural and anthropogenic aerosol on cloud base droplet size distributions in clouds over the South China Sea and Western Pacific"

We would like to thank the reviewer for providing comments and suggestions to this manuscript. Reviewer comments are in black text, responses to comments is in red text, and added text to the manuscript is in blue text.

Response to Reviewer

L465-6. This is not normalization. LWC within a size bin must be considerably less than LWC of the entire spectrum. Thus, concentrations in all bins are reduced by this "normalization." This reduction would be greater for the smaller size bins that would have less LWC due to smaller sizes to the third power. Multiplying by the ratio of LWC within each bin to the mean or median LWC of the spectrum would be a normalization. Or concentrations could be normalized according to the size widths of the bins? This needs explanation. Is whatever normalization that was or should be applied a common practice in cloud microphysics research?

The reviewer misunderstood the calculations as a result of our method not being clearly explained. We have changed the description and added an equation so that the method is now clear.

The following was added:

Figure 12 A shows the statistics of the normalized mass distribution function defined as

$$m_n(D) = \frac{\frac{\pi}{6}\rho_w n(D)D^3\Delta D}{\sum_D \frac{\pi}{6}\rho_w n(D)D^3\Delta D}$$

for all the droplet spectra in the marine category. The normalized mass distribution function was used to account for the fact that the aircraft sampled at different distances above cloud base and therefore encountered measured droplet spectra with different values of LWC. The remaining panels (Figs. 12 B-C) show the statistics of the normalized mass distribution functions for the other two aerosol source regions together with the marine spectra.

L475-6. This assertion requires explanation. Calculate the broader spectra and demonstrate. Is this over the entire size range or over some part of the droplet size range. Moreover, ship emissions are included in Fig. 12B not 12C, which is biomass. But even so panel B includes industrial. So, it is a further assertion to single out ships from industrial when both are included in Fig. 12B.

The sentence was removed.

L484-489. This paragraph is an out of context assertion. In order to stand it requires data backup and proper context.

The paragraph was removed.

L36-38. This is an assertion that was not demonstrated. Explain the meaning of Fig. 3. There are 10 plots in A & B. Are they related one-by-one to each other or is this just a coincidence? How are the two panels related? Apparently, you want to show that the cloud measurements were above the LCL.

This comment is related to figure 3, but the line number reference doesn't match. The reviewer is correct that we wanted to show the cloud measurements were at a range of altitude above the LCL. We added text to explain this.

The cloud penetration altitudes are shown in figure 3B. These figures together show that the cloud base penetrations used in this analysis occurred no more than 400 m above cloud base.

L287-291. How do the emissions of a research vessel compare to those of cargo and tanker vessels? I doubt that Sally Ride used Bunker Fuel. What fuel did it use?

The Sally Ride uses diesel fuel, but at a finer grade than the bunker fuel used by tanker vessels at the time of CAMP$^2$Ex. We made a note of this in the paper.

The R/V Sally Ride uses finer grade diesel fuel compared to the bunker fuel used by cargo and tanker ships at the time of CAMP$^2$Ex, although similar chemical components can be detected in the ship plumes from both fuel types.

L603. Four hours seems like a short time for a ship plume to disappear. Is there a reference to this fact?

We agree that four hours was an arbitrary time. We felt that beyond that time there would be too high of an uncertainty in the plume position. We chose the four-hour limit based on past work in Aliabadi et al. 2016. Reference was added to the paper.

Minor suggestions:

L30. Insert clean before marine.

Added clean

L33 & 35. SO4 should not be the same. Consistent with L350 & L513 SO4 should be 2.3 µg/m3 in L33. Also consistent with L350 & L513 ORG should be 2.2 µg/m3 in L32 and NH4 should be 0.3 µg/m3. The others in L32-3 are consistent with L349 & L350 and L523-4.

Fixed to be the same as the values in section 4

L43 & 45. Insert clean before marine.

Added clean

L46. Add d to influence.

Added

L63-64. Insert Hallett et al. (1989).

Reference added

L77. Insert Hudson et al. (2009) and Hudson & Noble (2014). I appreciate Nc. But now Nc can be employed in L80, L276, L402 and L455.

References added. $N_c$ changed throughout.

L81. Insert Hudson & Yum (2002).

Reference added

L86-7. Move constantly in front of into.

Changed

L107. Add Twohy et al. (2001).

Reference added

L290. Complementary.

Fixed.

L331. Change aerosols to particles.

Changed to particles

L339. Insert clean before marine.

Added

L401. Delete in length.

Deleted

L588. Remove away.

Removed